# Targeting the Electron Transport System for Enhanced Longevity

**DOI:** 10.3390/biom15050614

**Published:** 2025-04-23

**Authors:** Marko Radovic, Lucas P. Gartzke, Simon E. Wink, Joris A. van der Kleij, Frouwkje A. Politiek, Guido Krenning

**Affiliations:** Department of Clinical Pharmacy and Pharmacology, Section of Experimental Pharmacology, University Medical Center Groningen, University of Groningen, Hanzeplein 1 (AP50), 9713 GZ Groningen, The Netherlands; m.radovic@umcg.nl (M.R.); j.a.van.der.kleij@umcg.nl (J.A.v.d.K.); f.a.politiek@umcg.nl (F.A.P.)

**Keywords:** longevity, mitochondria, electron transport system, drug targeting, biomolecules

## Abstract

Damage to mitochondrial DNA (mtDNA) results in defective electron transport system (ETS) complexes, initiating a cycle of impaired oxidative phosphorylation (OXPHOS), increased reactive oxygen species (ROS) production, and chronic low-grade inflammation (inflammaging). This culminates in energy failure, cellular senescence, and progressive tissue degeneration. Rapamycin and metformin are the most extensively studied longevity drugs. Rapamycin inhibits mTORC1, promoting mitophagy, enhancing mitochondrial biogenesis, and reducing inflammation. Metformin partially inhibits Complex I, lowering reverse electron transfer (RET)-induced ROS formation and activating AMPK to stimulate autophagy and mitochondrial turnover. Both compounds mimic caloric restriction, shift metabolism toward a catabolic state, and confer preclinical—and, in the case of metformin, clinical—longevity benefits. More recently, small molecules directly targeting mitochondrial membranes and ETS components have emerged. Compounds such as Elamipretide, Sonlicromanol, SUL-138, and others modulate metabolism and mitochondrial function while exhibiting similarities to metformin and rapamycin, highlighting their potential in promoting longevity. The key question moving forward is whether these interventions should be applied chronically to sustain mitochondrial health or intermittently during episodes of stress. A pragmatic strategy may combine chronic metformin use with targeted mitochondrial therapies during acute physiological stress.

## 1. Aging: Mitochondrial Dysfunction at the Center

### 1.1. Aging Hallmarks

Aging is characterized by a continuous decline of physiologic function culminating in death [1]. In contrast, longevity relays the ability to live longer, thereby mitigating the effects of aging. Lifespan refers to the total duration of an individual’s life, encompassing both the effects of aging and interventions targeted to confer longevity. Currently, 12 hallmarks characterize aging [2], which—in an attempt to establish a sequential hierarchy of events—are further organized into the following 3 categories: primary, antagonistic, and integrative hallmarks. In this framework, primary hallmarks should be viewed as the root cause of aging, genomic instability, telomere attrition, epigenetic alterations, loss of proteostasis, and impaired macroautophagy. Conversely, antagonistic hallmarks, such as deregulated nutrient sensing, mitochondrial dysfunction, and senescence, manifest through the cellular response to primary hallmarks and, initially, can be protective but, over time, contribute to progressive aging. Integrative hallmarks arise as a consequence of primary and antagonistic hallmarks, consisting of stem cell exhaustion, altered intercellular communication, chronic (low-grade) inflammation, and dysbiosis [2]. Consequently, longevity is achieved by delaying, attenuating, or entirely avoiding the manifestation of aging hallmarks; as such, their absence serves as a defining feature of the longevity phenotype.

Due to the complex interplay of factors contributing to aging, untangling the definitive starting point remains challenging. Even so, genomic instability is widely regarded as a central component, exacerbating early vulnerabilities and propagating them into aging hallmarks [3,4]. Accumulated mutations caused by exogenous and endogenous oxidizing agents, as well as DNA replication and repair errors, all contribute to genomic instability, compromising the blueprint of cellular components and organelles [5,6]. These mutations jeopardize the production of cellular components and increase cellular waste [7,8]. Telomere attrition further contributes to this, as it limits the cellular replicative capacity, thereby contributing to tissue aging by slowing the influx of new cells and instead favoring senescence [9,10]. In keeping with this, epigenetic alterations lead to dysregulated gene expression, blunting the cell’s ability to respond to stimuli by transcribing and translating nuclear DNA into functional proteins and maintaining protein homeostasis [11,12]. The loss of proteostasis is further exacerbated not only by the damage or misassembly of proteins essential for cellular repair and maintenance but also by a broader decline in (macro)autophagy [13,14,15]. Over time the compounded effects of the primary aging hallmarks manifest in age-associated phenotypes and diseases. For example, genomic instability promotes the onset of cancer, while the epigenetically driven loss of proteostasis and impaired macroautophagy favor the accumulation of misfolded proteins, contributing to neurodegenerative diseases such as Alzheimer’s [16,17]. Notably, mitochondrial dysfunction has been identified as both a causal and modifiable factor in nearly all chronic diseases associated with aging [18,19,20,21]. While mitochondrial dysfunction in this framework can be viewed as a consequence of primary aging hallmarks, here, we challenge this view, and position mitochondrial dysfunction at the center of aging and the development of age-related pathologies.

### 1.2. The Aging Mitochondria

Mitochondria are highly complex organelles that provide critical cellular functions, including energy production and redox, amino acid, and metal ion homeostasis, thereby safeguarding proper cell functioning and affecting longevity [22,23]. Interestingly, all of the primary hallmarks of aging are directly applicable to the mitochondrion itself, as they exhibit genomic instability through mitochondrial DNA (mtDNA) damage, experience telomere-independent replication stress, and suffer from impaired proteostasis due to the accumulation of misfolded mitochondrial proteins [24,25,26]. This suggests that mitochondrial dysfunction is not merely a downstream consequence but an active driver of cellular aging and systemic decline. Here, perhaps the most relevant function of the mitochondria is to facilitate oxidative phosphorylation (OXPHOS), whereby the oxidation of nutrients is coupled with adenosine triphosphate (ATP) synthesis [27]. These processes are linked through a series of five protein complexes collectively comprising the electron transport system (ETS). Here, electrons derived from NADH via Complex I (NADH/ubiquinone oxidoreductase) and succinate via Complex II (succinate dehydrogenase) enter the ETS, where they are sequentially transferred through ubiquinone (UQ) to Complex III, then to cytochrome C, and, finally, to Complex IV, where molecular oxygen (O_2_) is reduced to water (H_2_O) [27]. Simultaneously, protons (H^+^) are translocated across the inner mitochondrial membrane at Complexes I, III, and IV, establishing the electrochemical gradient (proton motive force) necessary to drive ATP synthesis via Complex V (ATP synthase) [27]. However, this electron transfer is error-prone, and a small percentage of electrons leak from the ETS, prematurely reacting with oxygen and partially reducing it to form superoxide anions (O_2_^−^•), also known as the proximal oxygen radical [28]. These represent one of several reactive oxygen species (ROS) generated within mitochondria, alongside hydrogen peroxide (H_2_O_2_) and hydroxyl radicals (•OH), making these organelles the primary source of intracellular ROS [28]. Under physiological conditions, these radicals serve as signaling molecules, which are involved in redox homeostasis and adaptive stress responses [29]. However, when ROS production exceeds the cellular antioxidant capacity, they begin oxidizing surrounding proteins, lipids, and DNA, leading to the progressive accumulation of damaged mitochondrial components, the destabilization of mitochondrial membrane complexes, and increased mitochondrial membrane permeability. This, in turn, shifts the mitochondrial output away from ATP synthesis and instead favors the production of additional reactive species, including reactive nitrogen species (RNS) and lipid radicals. Over time, this creates a bioenergetic deficit, on the one hand, and disturbed mitochondrial homeostasis, on the other. The severity of this oxidative burden is further reflected in the fact that mitochondria retain their own genome, which remains highly vulnerable to ROS-induced damage. 

Most mitochondrial proteins are encoded by nuclear DNA as a result of evolutionary endosymbiotic gene transfer [30]. However, mitochondria still harbor multiple copies of their own circular mtDNA, which encodes 37 genes, including 13 essential components of the ETS—specifically, Complexes I, III, and IV, and ATP synthase [31]. Unlike nuclear DNA, mtDNA lacks protective histones, is located close to the radical-producing ETS, and has fewer repair mechanisms, rendering it highly susceptible to oxidative damage [32]. As a result, mtDNA mutations can further exacerbate ROS production and compromise OXPHOS function by encoding mutated components of the ETS, contributing to a self-reinforcing cycle of damage and energy deficiency. To counteract this, mitochondria operate as an interconnected reticulum, forming a dynamic network which preserves function through endogenous quality control mechanisms [33], whereby damaged components are selectively eliminated through reticulum dynamics [34]. Fusion/fission dynamics enable the redistribution of functional mtDNA and the removal of defective regions, maintaining mitochondrial integrity [35], while severely damaged mitochondria undergo complete degradation via mitochondrial autophagy (mitophagy) [36]. These mitochondrial dynamics tend to promote increasing homoplasmy, as they facilitate the exchange and removal of both healthy and damaged mtDNA [24]. Consequently, the combination of a limited repair capacity and reliance on quality control mechanisms facilitates the gradual accumulation of mtDNA mutations over time, and contributes to the decline in mitochondrial function as we age [32], which is particularly seen in high-energy-demanding organs [24]. Notably, this notion of mtDNA mutation accumulation alone as a primary driver of aging has been challenged and is a topic of debate [37,38,39]. Even so, while the incidence of mtDNA mutations may remain stable with age, their accumulation through clonal expansion has been observed to increase, progressively overburdening the non-mutated copies [24]. Highly pathogenic mtDNA mutations are likely rapidly eliminated through mitochondrial dynamics and selective pressures, whereas less pathogenic mutations may evade these mechanisms. These milder mutations accumulate over time via clonal expansion, surpassing their non-mutated counterparts and manifesting their deleterious effects when selective pressure is no longer sufficient to eliminate them. Consequently, ROS-induced protein damage and mutated mtDNA genes lead to the progressive loss in proteostasis, activating the mitochondrial unfolded protein response (UPRmt), a protective mechanism that attempts to restore mitochondrial proteostasis under stress conditions. However, while UPRmt activation is initially beneficial, its prolonged activity can confer a state of constant mitochondrial repair, inadvertently promoting the accumulation of mtDNA mutations, and possibly also contributing to the age-associated decline in function [26].

In parallel, chronic mitochondrial stress disrupts the balance between mitochondrial biogenesis and mitophagy, leading to a gradual reduction in mitochondrial mass (the mitochondrial copy number) with increasing biological age [40]. A key contributor to this decrease in mitochondrial maintenance is the reduced availability of NAD^+^, a critical cofactor required for sirtuin activation [41]. As the ETS efficiency decreases due to accumulated mtDNA damage and impaired subunit assembly, NAD^+^ recycling is impaired, suppressing sirtuin 1/3 activity and diminishing the activity of peroxisome proliferator-activated receptor gamma coactivator-1 alpha (PGC-1α), a key regulator of mitogenesis, thus limiting the production of new mitochondria [42]. This contributes to a state of deregulated mitochondrial substrate sensing, which mirrors the systemic deregulation of nutrient sensing observed with aging [43]. Concurrently, the removal of damaged mitochondria through mitophagy declines with age [44]. This is largely due to a reduction in PINK1/Parkin signaling caused by chronic mitochondrial depolarization, which stems from the compromised outer mitochondrial membrane integrity, preventing the accumulation of PINK1, and the subsequent activation of Parkin [45]. Additionally, metabolic aging blunts the cellular response to hormetic stress while promoting mTOR activation and deactivating AMPK, which inhibits autophagy and mitophagy downstream, further suppressing PINK1/Parkin activity [46,47]. Systemically, the increased permeability of the mitochondrial membrane favors the release of mitochondrial damage-associated molecular patterns (DAMPs), such as cell-free mtDNA, activating the inflammasome via NLRP3, causing chronic low-grade inflammation [48]. 

Together, this shows that mitochondrial aging and associated dysfunction is the central driver of chronic low-grade inflammation, i.e., inflammaging, which drives genomic instability through DNA oxidation and favors cellular senescence [49,50,51]. Below (Figure 1), we illustrate our perspective that aging starts inside the mitochondria with the slow but continuous ROS-driven oxidation of mtDNA and the faulty assembly of mitochondrial subunits, which, over time, leads to impaired oxidative phosphorylation (OXPHOS), i.e., mitochondrial dysfunction. Additionally, mitochondrial dynamics (fission and fusion) propagate the expansion of mtDNA lesions, leading to the clonal expansion of damaged mitochondria, thereby exacerbating mitochondrial dysfunction, leading to the release of mitochondrial DAMPs, such as cytochrome C and cell-free mtDNA, which favor inflammasome activation and low-grade inflammation. The chronic inflammatory signaling culminates in cellular senescence, whereas increased mitochondrial ROS leads to nuclear DNA oxidation, causing genomic instability and the subsequent disabling of macroautophagy, as well as the loss of cellular proteostasis, driving progressive tissue dysfunction. Given that mitochondrial function, particularly OXPHOS efficiency and ROS production, plays a central role in aging, this review will explore both established and emerging compounds that converge on mitochondrial pathways to promote longevity. We will first examine drugs with well-documented lifespan-extending effects, highlighting their connection to mitochondrial metabolism and OXPHOS, followed by a discussion of mitochondrially targeted compounds that may induce similar protective mechanisms.

## 2. Longevity Drugs Modulate the mTOR–AMPK Axis, Converging on the Mitochondrial Level

Several molecular mechanisms have been linked to longevity, and drugs targeting these confer longevity in the (pre)clinical arena. Two well-characterized drugs, i.e., metformin and rapamycin, target distinct pathways that influence aging. This section examines their mechanisms and their potential to increase lifespan. 

Metformin, also known as 1,1-dimethylbiguanide, is an oral anti-hyperglycemic and insulin-sensitizing drug. Since its introduction in the 1950s, when it was shown to have glucose-lowering effects in diabetic patients, it developed to become the gold standard in treating type 2 diabetes [52]. Over the years, metformin has also been increasingly prescribed off-label for pre-diabetes and polycystic ovary syndrome, and as an adjunct therapy in type 1 diabetes [53]. More recently, metformin has garnered interest in longevity research. Biguanides increased lifespan in preclinical models; more specifically, phenformin increased the mean lifespan of female C3H/Sn mice, while buformin increased the lifespan of aged female LIO rats. In female SHR mice, chronic metformin treatment extended the mean and maximum lifespan, highlighting its potential as a longevity-promoting drug [54]. Subsequent preclinical studies have repeated these experiments with varying success rates [55]. Findings from the NIH Interventions Testing Program demonstrated that metformin monotherapy did not promote longevity in UM-HET3 mice [56]. However, clinical evidence supports the idea that metformin increases lifespan, particularly in overweight individuals with diabetes mellitus type 2 (DMT2). Specifically, metformin-treated patients live longer than non-diabetic controls, despite their higher baseline weight and morbidity [57,58,59]. While short-term benefits are consistently observed in both diabetic and non-diabetic individuals, long-term advantages appear less certain, as the positive effects of metformin treatment diminish over time [60,61]. Expectedly, the uncertainty of the effects that this drug has on longevity resulted from clinical trials focused on longevity. One of the first being the Metformin in Longevity Study (MILES) [62,63], focusing on the effects that metformin has on changes in gene expression in aging adults. Their transcriptomic data suggest that metformin confers longevity by enhancing OXPHOS, specifically by stimulating beta-oxidation, and activating DNA repair, both governed by the upstream regulation of mTORC1. More recently, and still ongoing, the ‘Targeting Aging with Metformin’ (TAME) trial [64,65] is focusing on the effect that metformin has on delaying age-related chronic diseases and its effects on healthspan by tracking multiple aging biomarkers in non-diabetic adults.

Rapamycin, a natural macrocyclic lactone antibiotic, is an immunosuppressant widely used to prevent organ transplant rejection by inhibiting the mTOR (mechanistic target of rapamycin) pathway. Since its introduction in 1999, rapamycin has also gained prominence as a leading candidate for increasing longevity due to its potent effects on cellular aging effects. In UM-HET3 mice, rapamycin treatment extended the lifespan in both sexes [66,67]. Clinical data highlight rapamycin’s potential to reverse age-related immunosenescence, as pre-treated individuals contract fewer respiratory infections and respond better to influenza immunization [68,69]. Interestingly, metformin and rapamycin have synergistic effects on longevity; combined treatment with metformin and rapamycin starting at 9 months of age increases longevity [67,70]. Mechanistically, the positive effects of both rapamycin and metformin are reflected in the bioenergetic state that they induce. Rapamycin mimics starvation by deactivating mTORC1, thereby promoting catabolic processes and signaling a low-energy state. Metformin similarly lowers the cellular energy status by partially inhibiting Complex I, thereby reducing ATP levels while increasing adenosine monophosphate (AMP) [71]. Notably, beyond its AMPK-dependent effects, metformin also directly inhibits mTORC1 in a GTPase-dependent manner, further reinforcing its role in energy and growth regulation [72]. As such, both drugs mimic starvation and indirectly activate AMPK [73]. The starvation phenotype is also often induced by caloric restriction, a well-established and reproducible intervention for lifespan extension across various species, and is characterized by a catabolic switch to fatty acid oxidation [74]. Both rapamycin and metformin direct metabolism towards fatty acid β-oxidation, which—in the case of rapamycin—is coupled to increased mitochondrial biogenesis and function [75,76]. 

Despite the clearly demonstrated anti-aging effects of rapamycin, it is also established that it induces glucose intolerance [77,78,79]. Therefore, the synergistic effects of rapamycin and metformin can, at least in part, be explained by the blood glucose-lowering effect of concomitant metformin treatment. Recently, it has been shown that metformin’s blood glucose-lowering effect is dependent on the ND1 binding site of Complex I, promoting glycolysis to maintain cellular ATP [80,81,82]. Through increasing glycolysis, metformin favors NADH oxidation to NAD^+^ through lactate dehydrogenase (LDH), which converts pyruvate to lactate and oxidizes NADH back to NAD^+^ [83]. Additionally, through (indirect) AMPK activation, metformin activates NAD^+^ salvage pathways, such as nicotinamide phosphoribosyltransferase (NAMPT), thereby enhancing NAD^+^ synthesis from nicotinamide [84,85,86,87]. The resulting increased NAD^+^ favors sirtuin activation, promoting mitochondrial biogenesis through PGC-1α, and promoting the upregulation of ROS-detoxifying enzymes such as SOD2 [88,89]. Finally, metformin—just like rapamycin—directly inhibits mTORC1, thus explaining the overlapping positive effects [52,72,90]. Given the central role of Complex I in the mechanism of action of metformin and conduction, we will now dissect mitochondrial aging by moving towards compounds that modulate oxidative phosphorylation (Figure 2).

## 3. Modulating Oxidative Phosphorylation Activity to Convey Longevity

To improve electron transfer and maintain the proton gradient, targeted compounds have been developed. These therapies aim to enhance ETS function or substitute for deficiencies, particularly in Complexes I and IV, so to maintain energy production and limit oxidative stress in aging cells and tissues. Here, we review the potential of pharmacological interventions that favor mitochondrial health to halt or even reverse aging, and place mitochondrial (dys)function at the center of (accelerated) aging. 

### 3.1. Respiratory Complex I: A Master Regulator of Longevity

Complex I, NADH ubiquinone oxidoreductase, is the largest and functionally first enzyme complex of the electron transport system, localized within the inner mitochondrial membrane (IMM). Complex I serves as an ETS entry point, accepting electrons from NADH. As such, Complex I oxidizes NADH to NAD^+^, transferring two electrons down the ETS to ubiquinone (UQ), reducing it to ubiquinol (UQH_2_) [91,92,93]. Additionally, Complex I absorbs the spontaneous energy derived from the electron transfer (exergonic) to translocate protons (H^+^) from the mitochondrial matrix into the mitochondrial intermembrane space (IMS) (endergonic). For every electron pair transferred, four H^+^ are translocated into the IMS, making Complex I essential in establishing the proton motive force—consisting of the electrochemical and pH gradient—consumed by Complex V (ATP synthase) to generate ATP [94,95,96]. Furthermore, Complex I facilitates Na^+^/H^+^ exchange, transferring not only H^+^ ions to the IMS, but exchanging them for Na^+^ ions. This charge-neutral exchange helps to maintain the membrane potential and sustain the electrochemical gradient, even as the pH gradient is utilized during ATP synthesis [97,98,99].

Besides its role in redox reactions and the maintenance of the mitochondrial membrane potential, Complex I serves as a key regulator of energy status and redox sensing by its direct regulation of the NADH/NAD^+^ ratio. A high NADH/NAD^+^ ratio indicates high energy availability through the abundant presence of reducing agents, whereas a low NADH/NAD^+^ ratio signals energy shortage [100,101]. Complex I remains active (A-form) when high amounts of NADH are present, while switching to a less active, dormant state (D-form) in the absence of NADH. This dormant (D-form) state typically arises during conditions of hypoxia or ischemia, when NADH and ubiquinone levels are depleted, and represents a reversible transition from the active (A-form) state. While the D-form has been implicated in ischemia–reperfusion injury (IRI), its acute and reversible nature makes it less relevant to aging, which is characterized by chronic mitochondrial dysfunction, including sustained reductions in Complex I protein levels and persistent alterations in the NADH/NAD^+^ ratio [93,102,103]. The switch from the A- to D-state influences the formation of reactive oxygen species (ROS), allowing for electron transfer and partial reductions of O_2_ into superoxides (O_2_^−^) [28,104]. In its A-state, provided that sufficient oxidized UQ is present, Complex I facilitates forward electron transfer (FET), reduces UQ into UQH_2_, and drives OXPHOS. In the A-state, some ROS can form through an interaction with flavin mononucleotide (FMN) at the site of NADH oxidation [93,105,106]. Furthermore, UQ reduction can be facilitated through Complex II (succinate dehydrogenase or succinate-coenzyme Q reductase) and multiple other enzymes, depending on the substrate availability and the metabolic state of the cell [107,108,109]. An increase in the UQH_2_/UQ ratio creates a backlog of electrons within Complex I that increases ROS production through the increased electron transfer to O_2_. Moreover, in the presence of a high proton motive force, even in the absence of NADH, a reverse electron transfer (RET) from the reduced UQH_2_ pool can be facilitated on to Complex I, facilitating the production of superoxides at the UQ binding site of the Complex I ND1 subunit [93,110,111]. This is particularly evident in cases such as ischemia–reperfusion injury, where succinate, accumulated during ischemia, is rapidly oxidized by Complex II upon reperfusion, fully reducing the UQ pool and driving RET at Complex I [112,113,114]. This process is further exacerbated by the high proton motor force, which can result from impaired or inhibited Complex V (ATP synthase) activity [112]. Downstream, superoxides are reduced to H_2_O_2_ by the mitochondrial superoxide dismutase (SOD2), and, subsequently, H_2_O_2_ is converted to water by catalases [115]. Yet, in the presence of free Fe^2+^ and Cu^2+^ ions, H_2_O_2_ can form hydroxyl radicals through Fenton (like) reactions [116]. These and other radicals induce oxidative stress and damage to the surrounding molecules. Additionally, persistent RET-induced ROS formation can culminate in Complex I degradation to match the composition of the mitochondrial complexes to the substrate availability. In such cases, respiration becomes more Complex II- and Complex III-dependent.

Interestingly, metformin prevents the backlogging of electrons at Complex I through the partial inhibition of the ND1 binding site, limiting the conversion of NADH to NAD^+^ and preventing the (over)reduction of UQ, effectively reducing RET and RET-induced ROS formation [80,117,118]. Simultaneously, the metformin-induced reduction in Complex I activity decreases the proton motive force and lowers ATP production, as well as promotes glycolysis [118,119], thereby increasing the AMP/ATP ratio, which activates AMPK and protective downstream mechanisms, predominantly through promoting autophagy, PGC-1α-mediated mitochondrial biogenesis, and the NRF2-dependent upregulation of antioxidants [89,120,121]. Here, a key characteristic of metformin, though this remains a topic of debate, is its self-limiting inhibitory effect, in which it relies on the mitochondrial membrane potential to localize to Complex I. The critical distinction lies in metformin’s ability to lower the mitochondrial membrane potential, thereby reducing the electrochemical gradient and limiting its own accumulation at Complex I, resulting in the modulation rather than the complete inhibition of Complex I activity. In contrast, more potent Complex I inhibitors, such as rotenone, a highly hydrophobic small molecule, bind strongly to the UQ binding site independently from the membrane potential, and share mechanistic similarities with metformin [82,122]. Unlike metformin, rotenone is highly toxic and has been linked to neurodegenerative phenotypes, including Parkinsonism [123,124,125]. Interestingly, the inhibition of Complex I as a naturally occurring protective strategy is seen in maturing oocytes. Here, the endogenously regulated inhibition of Complex I formation was shown to preserve the ROS-free metabolism and maintain longevity [126]. This suggests that the physiological regulation of Complex I activity may be an adaptive mechanism for cellular protection, thus reinforcing the potential benefits of controlled OXPHOS modulation in aging and disease. Thus, metformin promotes longevity by the partial inhibition of Complex I and by evoking mild mitochondrial dysfunction, which increases the AMP/ATP ratio, and facilitates AMPK activation and its downstream beneficial effects. In terms of ROS, metformin produces low levels of ROS that favor mitochondrial biogenesis, yet prevents excessive ROS formation through persistent RET [127,128,129,130,131]. As such, the positive and documented effects of metformin treatment align with the hormesis theory, i.e., stress in low doses is beneficial.

Conversely, glitazones (thiazolidinediones, TDZs), another class of insulin-sensitizing drugs, which also have Complex I inhibitory effects, have disappointed in the clinical arena [132,133,134,135,136]. Beyond their well-characterized role in promoting mitochondrial biogenesis through the activation of PPARγ, glitazones are suggested to influence mitochondrial function by inhibiting Complex I and Complex III activity by up to 50% [132,137]. This inhibition appears to involve an interaction with a specific mitochondrial binding site, mitoNEET, an outer mitochondrial membrane protein that regulates the mitochondrial iron content and facilitates redox reactions involving a 2Fe-2S cluster [138,139,140,141,142]. Moreover, TDZs can reduce ROS production and increase antioxidant enzymes such as SOD [143,144,145]. Even so, clinical outcomes have been far from favorable, with multiple studies suggesting increased risks of heart failure and myocardial infarctions in elderly patients by the use of TDZs [134,135,136]. The underlying cause of these outcomes has been a topic of debate, attributing it to substrate overload in cardiac cells [146,147,148]. While TDZs share many of the beneficial effects of metformin, such as insulin sensitivity, glucose uptake, reduced Complex I activity, and anti-inflammatory effects, it remains enigmatic why their clinical effects differ so much [132,133,134,135,136,149]. What is clear is that modulating OXPHOS, while risky depending on inhibitor potency and off-target effects, presents a potential platform for anti-aging drugs. Therefore, in the remainder of this review, we will explore compounds that modulate OXPHOS and mitochondrial metabolism, and which have commonalities with metformin in the pathways that they affect (Figure 3).

### 3.2. NAD^+^ Boosters: NMN and NR

One strategy to mimic the longevity effects of metformin and rapamycin could involve lowering the NADH/NAD^+^ ratio. The positive effects of increased NAD^+^, along with the observation of some studies that linked aging to lower levels of NAD^+^, has sparked research interests into the effects of exogenous NAD^+^ supplementation to compensate for age-related energy disturbances to boost longevity [150]. In this context, the following two compounds have been extensively studied: nicotinamide mononucleotide (NMN) and nicotinamide riboside (NR). NMN is a precursor of NAD^+^ and is enzymatically converted to NAD^+^ in a one-step reaction by nicotinamide mononucleotide adenylyltransferase (NMNAT), thereby increasing the cellular NAD^+^ pool [151]. NR, on the other hand, can be utilized in the following two ways: (1) it can be converted by nicotinamide riboside kinase (NRK) NMN, after which NMNAT converts it to NAD^+^; (2) NR can be phosphorylated by purine nucleoside phosphorylase (PNP) to form nicotinamide (NAM), which enters the NAD^+^ salvage pathway and is converted to NAD^+^ by nicotinamide phosphoribosyltransferase (NAMPT) [152]. 

To dissect the efficacy of NMN and NR in conferring longevity, the relationship between NAD^+^ and aging must first be established. NAD^+^ is a well-established cofactor in sirtuin activation. As discussed previously, sirtuins activate PGC-1α, which promotes mitochondrial biogenesis, and could be beneficial to counteract the lowered mitochondrial mass observed with aging [153,154]. Additionally, sirtuins suppress inflammasome assembly and activation, and, as such, it seems plausible that NAD^+^ supplementation could dampen inflammation [88]. In parallel, NAD^+^ supplementation may facilitate DNA repair. Poly(ADP-ribose) polymerases (PARPs) are a family of enzymes involved in the DNA damage response (DDR) and serve to maintain genomic stability or facilitate cell death—predominantly parthanatos—in the case of an unrepairable DNA lesion. PARPs consume NAD^+^ to recruit other DDR proteins to the DNA damage site [155]. Hence, it is conceivable that increased NAD^+^ availability may foster increased DNA repair. 

Currently, there are no data available to discuss the use of NR or NMN in model organisms beyond yeast to confer longevity [156,157,158], yet the theoretical framework warrants further exploration. However, it is worth mentioning that NAD^+^ augmentation has been shown to extend lifespan and restore mitophagy in models of Werner syndrome—an accelerated aging disorder—in both *C. elegans* and Drosophila, suggesting that NAD^+^ repletion does counteract specific progeroid phenotypes and may prove effective against aging in future studies [159]. It is pertinent to highlight that studies disagree on whether aging is actually associated with reduced levels of NAD^+^ [150]. Interestingly, while NAD^+^ boosting confers longevity in yeast, it is unclear whether aging in yeast coincides with reduced NAD^+^ levels, as studies on this have produced conflicting results [157,160,161,162]. Contrary, in studies of C. elegans, mice and rats showed reduced levels of NAD^+^ with increasing age [163,164,165,166,167]. The majority of studies conducted in mice point towards reduced NAD^+^ levels with increasing age; however, it is of note that studies conducted in the same tissue, e.g., the brain, have produced conflicting results [150,168,169,170,171,172]. Finally, studies in (the tissues of) humans predominantly confirm decreased NAD^+^ levels with increasing age; in the case of the brain tissue, in vivo magnetic resonance studies suggest increased NADH/NAD^+^ ratios, indicating reduced NAD^+^ availability with increasing age [150,173,174]. The inherent difficulty in obtaining tissue from essential human parenchyma naturally prohibits studying this in human brains, and the difficulty in standardizing human experiments warrant a careful interpretation of the results. 

Regardless, the final verdict on NAD^+^ levels in aging is yet to be seen due to the lack of longevity data in more complex organisms than yeast. While a theoretical framework supporting supplementation with NAD^+^ precursors is convincing, clinical studies so far have only been able to demonstrate mild physical performance increases (e.g., increased walking distance in peripheral artery disease). NR supplementation conferred a mild improvement of Parkinson’s symptoms, but only when the analysis was corrected for NR responders [175]. NR therapy in chronic kidney disease had no effect on eGFR, and even decreased submaximal VO_2_, but did lower triglycerides [176]. As such, the perceived benefits of NAD^+^ supplementation through NMN or NR so far may thus be paralleled by the supplementation of antioxidants, namely, that the decreased levels of NAD^+^ and antioxidants are a symptom of aging but not a cause. Hence, simple supplementation is insufficient to modify aging outcomes. 

## 4. Redox Stress: (Targeted) Antioxidants and Senolytics

### 4.1. (Targeted) Antioxidants

Facilitating ROS-free respiration, a hallmark of longevity-associated compounds, can theoretically be achieved through direct ROS scavenging via antioxidants. However, despite this premise, antioxidants such as vitamin C and vitamin E have consistently failed to demonstrate significant health or lifespan-extending benefits in clinical trials, a conclusion supported by numerous reviews [177,178,179,180]. One key limitation is that highly reactive species, such as hydroxyl radicals (OH•), react instantaneously with surrounding molecules, making scavengers ineffective at sufficiently high local concentrations [179]. Given that antioxidants struggle with local bioavailability, mitochondria-targeted compounds such as MitoQ and MitoVitE attempt to overcome this limitation by selectively accumulating within mitochondria [181]. This is achieved with the addition of the lipophilic triphenylphosphonium (TPP) cation, which enables rapid uptake across the membranes towards the mitochondria, driven by the high mitochondrial membrane potential [182]. In MitoVitE, the TPP cation is conjugated to α-tocopherol (vitamin E), whereas, in MitoQ, it is attached to coenzyme Q10, both of which are well known for their antioxidant capacity [182,183]. Some studies have shown that MitoQ and MitoVitE exhibit superior antioxidant capacities compared to Trolox and vitamin E [183,184]. However, for MitoVitE, other studies found no significant differences in the efficacy compared to its non-targeted counterpart, α-tocopherol [185]. Moreover, MitoVitE has failed to prolong neuronal survival in rats following acute perinatal hypoxia–ischemia, and, at higher concentrations, has even exhibited neurotoxic effects [186,187,188]. In contrast to MitoVitE, MitoQ has shown more promising preclinical results, exhibiting protection against the loss of heart function, tissue damage, and mitochondrial dysfunction compared to TPMP or short-chain quinol controls in hearts subjected to ischemia–reperfusion injury isolated from treated rats [189]. However, MitoQ, in a clinical trial on Parkinson’s disease patients, showed no difference between the treated and placebo groups, suggesting either less involvement of oxidative stress in the disease pathophysiology or, similarly to that of non-targeted antioxidants, ROS overwhelming their antioxidant capacity—once again calling into question the viability of ROS scavenging as a therapeutic strategy [190]. 

Alternatively, targeting the mechanisms of ROS formation, such as reducing H_2_O_2_ production with superoxide dismutase (SOD)-like enzymes, rather than scavenging already-formed radicals with small molecules, may offer an alternative approach to prevent ROS-mediated damage and aging. An example is the case of mitoTEMPO, another drug developed to specifically localize to the mitochondria, utilizing the same TPP cation as MitoQ and MitoVitE, where it mimics and facilitates SOD function. Preclinical mice models of type 1 and 2 diabetes treated with mitoTEMPO showed reduced ROS generation, decreased apoptosis, and, ultimately, reduced myocardial hypertrophy [191]. It was also shown to improve blastocyst development in preimplantation porcine embryos by reducing mitochondrial superoxide levels, restoring the mitochondrial membrane potential, and increasing ATP production [192]. In fact, several other studies show similar protective effects on mitochondrial function in various age-associated disease models [193,194,195]. Currently, the first clinical trial for mitoTEMPO is underway (ClinicalTrials.gov ID NCT06424756) [196]. 

However, despite these effects in the pathological models of these targeted antioxidants, studies investigating their role in longevity remain limited and have demonstrated no effect. For instance, MitoQ failed to extend the lifespan of wild-type fruit flies, suggesting that its benefits may be restricted to contexts involving compromised or overwhelmed antioxidant defenses [197]. Similarly, in another study on mice, the MitoQ treatment did not affect lifespan [198]. If the free radical theory of aging holds true, such antioxidant benefits should translate into mitigating aging-related phenotypes and extend normal lifespan; yet, until today, the evidence for longevity effects is lacking. In fact, ROS may serve a more important role in signaling, and even more so in the selective pressure of cells, as we will elucidate next. 

### 4.2. Sonlicromanol (KH176)

In contrast to direct antioxidant strategies, an alternative approach focuses on regulating ROS signaling pathways and exploiting endogenous antioxidant defenses to facilitate ROS-free respiration while preserving essential mitochondrial functions. This strategy is exploited by Sonlicromanol, also known as KH176, a redox-modulating compound designed to restore mitochondrial function by reducing oxidative stress induced by dysfunctional Complex I activity, while at the same time preserving essential redox signaling [199,200,201]. Sonlicromanol selectively regulates the endogenous thioredoxin/peroxiredoxin (Trx/Prx) system, which is crucial for mitochondrial redox homeostasis [199]. Prx acts as peroxide scavengers, reducing hydrogen peroxide (H_2_O_2_) and lipid peroxides, while Trx recycles oxidized Prxs back to their active form by using electrons from NADPH via thioredoxin reductase (TrxR) [202]. Here, the active form of Sonlicromanol, being its quinone metabolite KH176m, seems to interact with Prx, increasing its peroxidase activity, converting it back into its active (reduced) form, and restoring the capacity for H_2_O_2_ scavenging [199]. Additionally, KH176m has been shown to selectively inhibit prostaglandin E_2_ (PGE_2_) biosynthesis by targeting microsomal prostaglandin E synthase-1 (mPGES-1) [203]. This mechanism is particularly relevant, as PGE_2_, a key immune-modulating lipid, has been shown to induce senescence via the reduction in OXPHOS and mitophagy, thereby promoting the accumulation of damaged mitochondria [204]. In a myocardial ischemia–reperfusion injury model in rats, Sonlicromanol exhibited cardioprotective effects by upregulating AMPK and eNOS expression, a response that may stem from its known redox-modulating activity, or which may indicate an additional direct pathway of activation that remains to be elucidated [205]. Nevertheless, by simultaneously enhancing mitochondrial antioxidant defenses and mitigating PGE_2_-driven senescence, KH176m emerges as a promising therapeutic candidate for combating mitochondrial dysfunction and inflammation-associated aging. Moreover, beyond its direct effects on ROS signaling, Sonlicromanol is particularly intriguing due to its regulation of peroxiredoxin, whose role is well-established in modulating longevity across various model organisms, adding another layer of significance to Sonlicromanol as a longevity-promoting compound [202]. To date, Sonlicromanol has been tested in phase I and II clinical trials, demonstrating good safety and tolerability, with the next phase III trial (NCT06451757), focusing on primary mitochondrial diseases, currently awaiting initiation [200,206,207].

### 4.3. Procyanidin C1 (PCC1)

Although seemingly counterintuitive, there is a growing interest in exploiting ROS to facilitate the clearance of damaged mitochondria and to promote the biogenesis of functional mitochondria as an emerging pharmacological strategy. This concept is exemplified by PCC1, a senolytic compound that selectively targets senescent mitochondria to promote their clearance via ROS-mediated mechanisms. Here, the dual role of ROS—as both damaging agents and mediators of repair—underscores the complexity of mitochondrial regulation and involvement in longevity. Above, we discussed how mitochondrial dysfunction drives senescence, an integrative hallmark of aging. While senescence is a terminal end-point for an individual cell, in concert, senescent cells act to promote inflammaging through the senescence-associated secretory phenotype (SASP), which promotes surrounding cells to become senescent as well, thus accelerating aging [208].

An intriguing approach to extending longevity is the use of compounds that selectively remove senescent cells, i.e., senolytics [209]. Procyanidin 1 (PCC1) is a senolytic compound that affects mitochondrial function in senescent cells to prolong lifespan, while also significantly reducing SASP factors and senescence markers [210]. Unlike other procyanidins, which primarily act through broad antioxidant activity or the modulation of inflammatory pathways like NF-κB, MAPK, and Nrf2, PCC1 directly induces mitochondrial dysfunction in senescent cells, exploiting the inherent vulnerability of these cells and clearing them though mitochondrially driven apoptosis [210]. This is supported by the elevated ROS levels, suggesting that PCC1 disrupts mitochondrial homeostasis [210]. These effects are also mitigated by the antioxidant HS-1793, while being exacerbated by mitochondrial disruptors such as CCCP [210]. This positions ROS as a potential tool to counteract aging when strategically modulated. 

PCC1 induces mitochondrial membrane potential disturbances, leading to cytochrome C release, and ultimately triggering apoptosis and the clearance of senescent cells [210,211]. Moreover, a recent study using single-cell RNA sequencing, comparing untreated and PCC1-treated cells from aged mice retina, has shown that treatment with PCC1 led to the downregulation of genes with reported roles in oxidative stress response, while upregulating NADH regeneration pathways [212]. While not explicitly tested, it is plausible that the increase in NADH regeneration pathways indicates a reduced NADH/NAD^+^ ratio, which could trigger the mito-protective effects of SIRT3 [213]. In concordance, PCC1 was shown to increase SIRT3 activity in an acidic pH stress model [214]. While it remains challenging to merge conclusions from different stress models, it seems that PCC1 exploits the inherent vulnerabilities of senescent cells, which harbor more dysfunctional mitochondria and, as such, are more prone to ROS-mediated damage than proliferating cells. Hence, this suggests that exacerbating mitochondrial stress in senescent cells leads to their clearance [210], while simultaneously promoting mitochondrial biogenesis through the induction of SIRT3 in proliferating cells, potentially halting or even reversing aging (Figure 4).

## 5. Oxidative Phosphorylation: Targeting the Respiratory Complexes

### 5.1. The Respiratory Complex III–Cytochrome C–Complex IV Supercomplex

Another promising avenue for pharmacological intervention targeting not only ROS production but also the mitochondrial-induced inflammatory state is the Complex III–cytochrome C (Cyt C)–Complex IV supercomplex [215]. Reduced ubiquinol (UQH_2_) relies on Complex III to transfer electrons to Cyt C, which subsequently transfers electrons to Complex IV, where the complete reduction of oxygen (O_2_) to water (H_2_O) is facilitated [27]. At the same time, Complex III and Complex IV contribute to the formation of the proton-motive force (PMF) by translocating protons (H^+^) across the inner mitochondrial membrane (IMM). Unlike Complex III and Complex IV, which are membrane-bound, Cyt C is a soluble protein of the mitochondrial intermembrane space (IMS) [27]. Here, the highly cationic Cyt C is maintained in close proximity to the two protein complexes through electrostatic interactions with the anionic lipid cardiolipin (CL) [216]. CLs are lipids specifically localized in the IMM, where—during homeostasis—they contribute to the stability of membrane proteins, facilitate protein–protein interactions, and promote the formation of the Complex III–Cyt C–Complex IV supercomplex [215,216]. However, mitochondrial dysfunction, including the loss of mitochondrial membrane potential (ΔΨm), reduced adenosine triphosphate (ATP) production, increased ROS production, and altered calcium (Ca^2^^+^) homeostasis, disrupts the Cyt C–CL interaction [217,218,219]. This shift changes the interaction from a weakly bound electrostatic association to a tightly bound hydrophobic interaction. Although the precise mechanism underlying this transition remains unclear, the outcome is a conformational unfolding of Cyt C, exposing its heme group, which facilitates undesired redox reactions. This converts Cyt C into a peroxidase capable of generating hydroperoxides and lipid peroxyl radicals (LOO•), thereby propagating lipid peroxidation [217]. This peroxidation cycle further impairs the ETS, exacerbating mitochondrial damage and increasing ROS output. These oxidative processes initiate mitochondrial stress responses, leading to the translocation of CL to the outer mitochondrial membrane (OMM). At the OMM, CL serves as a platform for NLR family pyrin domain-containing 3 (NLRP3) inflammasome assembly, which, in turn, activates the release of pro-inflammatory cytokines, such as interleukin-1β (IL-1β) and interleukin-18 (IL-18) [220]. This inflammatory signaling contributes to a broader cellular stress response and is further exacerbated by the release of mitochondrial damage-associated molecular patterns (DAMPs), such as mtDNA, ATP, ROS, cardiolipin, and Cyt C itself [221]. While not entirely clear what causes their release, mitochondrial DAMPs, recognized by pattern recognition receptors (PRRs), amplify inflammatory cascades and drive chronic inflammation [221,222,223]. Under normal conditions, dysfunctional mitochondria are selectively degraded via mitophagy; however, as already mentioned, aging impairs mitophagic clearance, allowing damaged mitochondria and their inflammatory signals to persist [44,224]. Impaired mitochondrial quality control can lead to both mPTP opening and outer mitochondrial membrane permeabilization (MOMP), resulting in the sustained release of DAMPs into the surrounding tissues, thereby propagating inflammation and, ultimately, contributing to systemic inflammaging [222,225,226,227,228,229]. Additionally, mitochondrial permeability increases with age, not only disrupting the oxidative phosphorylation (OXPHOS) efficiency but also amplifying pro-apoptotic and inflammatory cascades [230], thus further exacerbating age-related tissue dysfunction. Given the central role of the Complex III–Cyt C–Complex IV supercomplex in maintaining mitochondrial integrity and regulating inflammation, thus preserving its stability, in the following sections we will explore pharmacological compounds that may counteract these effects by stabilizing mitochondrial supercomplexes and optimizing OXPHOS efficiency, thereby reducing mitochondrial permeabilization, dampening inflammation, and potentially delaying inflammaging.

### 5.2. Elamipretide (SS-31, MTP-131, and Bendavia)

Elamipretide is a first-in-class compound claimed to prevent Cyt C peroxidase transition and oxidation of cardiolipin (CL) [231]. Having a higher affinity for CL, when compared to other lipids of the IMM, Elamipretide was shown to prevent the hydrophobic interaction between CL and Cyt C [232]. In doing so, Elamipretide prevents the unfolding of Cyt C and the exposure of the heme group, precluding peroxidase activity while preserving the ability to reduce Cyt C through GSH and vitamin C in the presence of CL [232]. Moreover, Elamipretide was shown to reverse the disrupted CL–Cyt C interaction observed during mitochondrial isolations, thereby restoring the native conformation of Cyt C. This preservation supports the electron transfer between Complex III and Complex IV, ultimately leading to increased respiration and ATP production. Moreover, Elamipretide by itself has no effect on the reduction in Cyt C, thus suggesting that Elamipretide, to be effective, requires unfolded Cyt C. Indeed, Elamipretide preferentially interacts with the unfolded state of Cyt C, facilitating its reduction only when CL-induced structural changes expose critical redox-active sites [232]. In a more recent TAZ knockdown model, through which cardiolipin synthesis was impaired, Elamipretide slightly improved the supercomplex formation, reduced LEAK, and enhanced OXPHOS. However, in the absence of completely formed cardiolipin species, it failed to restore the physiological OXPHOS capacity and prevent cytochrome C efflux. This suggests that mature cardiolipin is required for Elamipretide to exert its protective effects [233]. While the exact mechanism of how Elamipretide preserves the electrostatic interaction between CL and Cyt C remains elusive, clinical trials have reported mixed effects; while cardiac cell apoptosis was reduced by Elamipretide, a single intravenous infusion of Elamipretide did not evoke therapeutic benefit in heart failure patients, suggesting that either a different pharmacotherapeutic regimen is needed or that the therapeutic blocking of the Cyt C peroxidase transition might not be enough to improve the clinical outcomes in age-related (cardiac) diseases. Nevertheless, increased resistance to cardiolipin peroxidation and mitochondria-induced apoptosis remain relevant factors in the context of lifespan extension. Notably, melatonin, an endogenous circadian rhythm regulator with comparable effects on cardiolipin (CL) peroxidation and cytochrome C release, has been shown to extend lifespan in both wild-type and senescent-accelerated mice, as well as other model organisms [234,235]. Given these parallels, Elamipretide warrants further investigation in this specific context—not only as a potential longevity-promoting compound but also as a tool to elucidate the role of the CL–Cyt C interaction in lifespan regulation (Figure 5).

### 5.3. Methylene Blue

Methylene Blue (MB) is a redox-active drug which was suggested to act as an alternative electron carrier by directly oxidizing NADH and subsequently donating electrons to Cyt C/Complex IV [236]. As a consequence, MB effectively circumvents the need for Complex I and III activity for energy production, thus preventing metabolic stalling, which would otherwise accumulate as the NADH/NAD^+^ ratio increases [237]. Moreover, recent studies have shown that the inhibition and even removal of any ETS component, including Complex IV, is mitigated by the addition of MB, suggesting that it might have the capacity to substitute the respiratory chain in its entirety [238]. The model suggests that MB undergoes a reversible redox cycling process, in which it is reduced to leucomethylene blue (LMB) by cellular reductants like NADH. It is then directly re-oxidized by molecular oxygen, leading to increased oxygen consumption that is independent of the standard oxidative phosphorylation pathways. By facilitating electron transfer away from NADH, MB prevents NADH accumulation and ensures the continuous regeneration of NAD^+^, thus sustaining the cellular metabolism, particularly under conditions of mitochondrial stress. Indeed, MB has been shown to sustain ATP production via substrate-level phosphorylation in the Krebs cycle, even when mitochondrial ATP synthesis is impaired [238]. This compensatory mechanism provides energy support in cases where traditional oxidative phosphorylation is compromised. However, since molecular oxygen serves as the electron acceptor in the oxidation of LMB, the final product appears to be H_2_O_2_, suggesting that dismutation by catalase or other peroxidases is essential for this compound to facilitate its function without compromising the surrounding molecules. Indeed, these enzymes are believed to be present at sufficiently high levels, thus minimizing the risk of peroxide accumulation [239]. Moreover, MB’s ability to sustain ATP production, even with compromised ETS activity, and to maintain NADH/NAD^+^ balance positions it as a potential anti-aging intervention by preserving cellular energy homeostasis and mitigating age-related metabolic decline [240]. Interestingly, MB seems to upregulate the cellular antioxidant defenses, and was shown to mitigate skin aging in both cultured cells and 3D-reconstructed human skin epidermis [241]. Similar observations were seen in mouse models of tauopathy, in which MB had neuroprotective effects by reducing the tau pathology and upregulating the expression of genes regulated by the NF-E2-related factor 2 (Nrf2) and antioxidant response element (ARE) [242]. Moreover, various other preclinical trials have shown similar beneficial effects of MB on age-related diseases, and, in particular, neurodegenerative effects [240]. However, in a preclinical study on aging mice, MB did not show significant improvements on age-associated bone loss, possibly suggesting that substituting the ETS is not enough to prevent its implication in aging [243]. Moreover, multiple clinical trials in Alzheimer’s disease patients have shown no significant improvement in disease progression [244,245,246]. While this raises concerns about MB’s potential as an anti-aging candidate, its extensive research history and FDA approval provide a foundation for future investigations into its effects on aging, particularly in less severe conditions compared to neurodegenerative diseases (Figure 6).

### 5.4. SUL-138 (SUL-109/238)

SUL compounds represent a first-in-class set of molecules with apparent effects on the Complex III–Cyt C–Complex IV supercomplex. SUL-109 was shown to increase the activity of not just Complex IV, but Complex I as well, outcompeting the effects of inhibitors such as rotenone and cyanide, and seemingly affecting the whole ETS [247]. In both in vitro and in vivo models of mitochondrial dysfunction, SUL compounds effectively maintained ATP production and precluded ROS formation [248,249,250,251]. Mechanistically, the more advanced SUL-138, the purified S-enantiomer of SUL-109, shifted the mitochondria towards beta-oxidation, effectively enhancing the pathways previously associated with prolonged lifespan and the caloric-restriction metabolism shift [74,252,253]. In line with these metabolic benefits, SUL-138 has also been shown to improve cognitive performance in both Alzheimer’s disease model mice (APP/PS1) and in healthy control mice, suggesting its potential as a neuroprotective agent, potentially conferring age-related benefits as opposed to just disease-specific benefits. Moreover, the administration of SUL-238, the HCl salt of SUL-138, in Ercc1-deficient mice, in a model of accelerated aging and DNA repair deficiency, was shown to reduce vascular aging by a yet-to-be-determined mechanism [254]. While certainly promising in terms of longevity and aging, the exact mechanism through which this 6-chromanol derivative works is yet to be determined. A phase I clinical trial (NCT06277492) aiming to establish safety and tolerability in healthy volunteers is currently ongoing, and the results should provide more clarity on whether targeting the Complex III–Cyt C–Complex IV supercomplex is feasible [255].

## 6. Future Directions for Mitochondrial Anti-Aging Drugs

Mitochondrial dysfunction is increasingly recognized as a central driver of aging and age-related disease. At the heart of this process lies the ETS, where dysregulated OXPHOS promotes excessive ROS formation, mtDNA damage, and cellular senescence, culminating in inflammaging [24,25,26,49,50,51]. Pharmacological interventions that modulate mitochondrial function, particularly at Complex I, have emerged as promising strategies to delay aging and extend healthspan. Two of the best-characterized longevity drugs, metformin and rapamycin, exert their effects through distinct upstream mechanisms, yet converge on the mitochondrial metabolism. (1) Metformin, a partial Complex I inhibitor, reduces reverse electron transfer (RET)-induced ROS production at the ND1 subunit, thus preserving mitochondrial integrity. It also induces a state of mild mitochondrial dysfunction that paradoxically favors longevity by activating AMPK, enhancing autophagy, promoting NAD^+^ salvage pathways, and stimulating mitochondrial biogenesis via PGC-1α. Importantly, its self-limiting action at Complex I prevents excessive inhibition, making metformin a very safe drug. Life-long metformin use is safe in both preclinical and clinical studies, with clinically apparent benefits in diabetic and obese patients [80,84,85,86,87,89,117,118,120,121,127,128,129,130,131]. (2) Rapamycin, in contrast, inhibits mTORC1, enhancing mitophagy and mitochondrial turnover, while shifting metabolism toward fatty acid β-oxidation. Despite robust evidence of lifespan extension in animal models, rapamycin’s more complex side-effect profile currently limits its feasibility as a chronic anti-aging therapy in humans, whereas intermittent regimens may, in the future, prove sustainable and effective [66,67,68,69,75,77,78,79].

Beyond these well-characterized agents, a new generation of small molecules directly targeting OXPHOS is emerging as a promising strategy to restore mitochondrial function and halt aging. Compounds such as Elamipretide and other 6-chromanol derivatives act by stabilizing ETS (super)complexes, preserving or restoring the mitochondrial membrane potential and limiting DAMP release [232,233,247,248,249,256,257,258]. Importantly, these compounds work in acute and chronic models of mitochondrial dysfunction, but remain to be evaluated for their longevity effects. This dual efficacy across acute and chronic disease models raises the following important question regarding their clinical application: should these compounds be administered continuously to maintain mitochondrial health and delay age-related decline, or should they be deployed intermittently during predictable episodes of mitochondrial stress, such as infection, surgery, or intensive care admissions?

Finally, mitochondrial dysfunction can also be positively leveraged for longevity. PCC1, a senolytic derived from procyanidins, offers a complementary approach by selectively clearing senescent cells. It does so by exacerbating mitochondrial dysfunction within these cells, thereby triggering apoptosis. In preclinical models, PCC1 reduces the pro-inflammatory secretory phenotype associated with cellular senescence, thereby mitigating chronic inflammation (inflammaging) and promoting tissue rejuvenation. This suggests the following dual benefit: reversing age-related tissue damage while extending healthspan [209,210,211,212].

The key question moving forward is whether these compounds should be implemented in a daily regimen to sustain mitochondrial health or administered intermittently during foreseeable stress. Striking the right balance between maintaining mitochondrial efficiency and harnessing the adaptive benefits of intermittent stress will be crucial for the next generation of anti-aging therapies. For now, a pragmatic strategy may involve chronic metformin use to preserve baseline mitochondrial function, combined with targeted mitochondrial interventions during acute physiological stress. Experimental validation of this approach is warranted. Preclinical models of ischemia–reperfusion, hypoxia–reoxygenation, cooling–rewarming, and sepsis reliably induce mitochondrial dysfunction and offer valuable platforms to test whether chronic metformin treatment, supplemented with acute small-molecule interventions during episodes of stress, can enhance the resilience and extend the healthspan.

## 7. Conclusions 

Mitochondrial function intertwines with lifespan wherein malfunctioning of the mitochondrial ETS particularly affects ageing processes and shortens lifespan. Pharmacological agents that mitigate mitochondrial dysfunction are in development and show promising effects in the preclinical arena by limiting ageing-related processes and modifying outcomes in age-related disease phenotypes. If these mitochondrial active anti-ageing drugs are also effective in humans is currently under clinical investigation. 

## Figures and Tables

**Figure 1 biomolecules-15-00614-f001:**
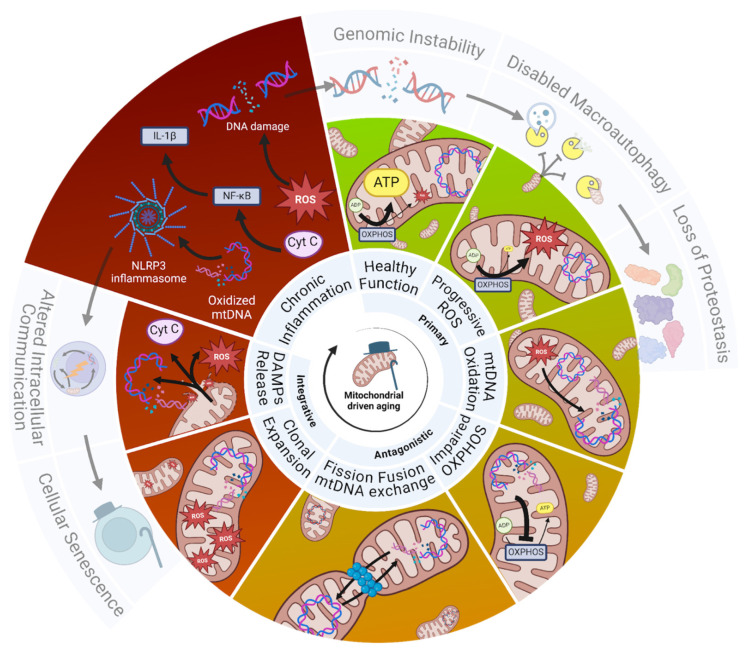
Mitochondrial hallmarks of aging. Mitochondrial dysfunction is the initiating event in aging. Increased ROS-induced damage to mitochondrial DNA (mtDNA) leads to the production of faulty electron transport chain (ETC) complexes, causing mitochondrial dysfunction. As a result, oxidative phosphorylation is impaired, reducing the ATP yield. Over time, the nature of mitochondrial dynamics promotes the exchange of damaged mtDNA, which propagates mutations through clonal expansion. This further exacerbates mitochondrial dysfunction, progressively shifting mitochondrial output from ATP production to increased ROS production. This, in turn, increases mitochondrial membrane permeability (MMP), triggering the release of mitochondrial damage-associated molecular patterns (DAMPs) and ROS. These signals drive systemic inflammation and cellular senescence. Simultaneously, mitochondrial dysfunction promotes nuclear DNA damage and genomic instability, culminating in the loss of proteostasis. Together, these processes establish a feedback loop, positioning mitochondrial dysfunction at the core of aging.

**Figure 2 biomolecules-15-00614-f002:**
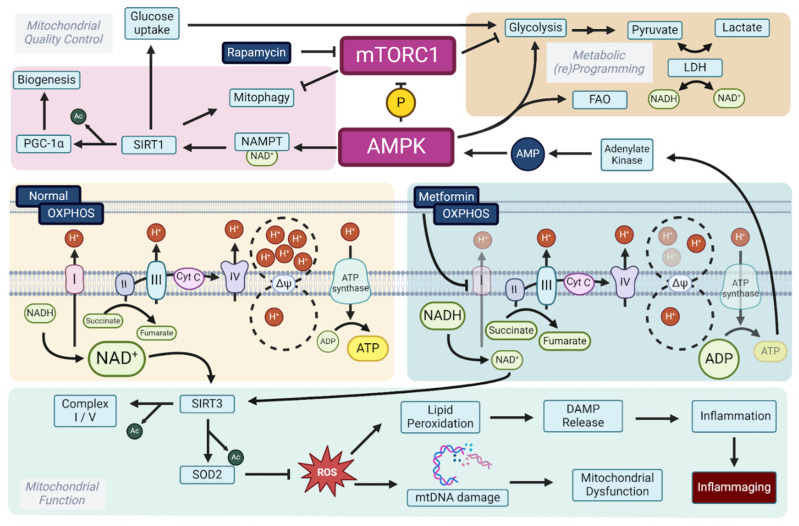
Metabolic and mitochondrial mechanisms of rapamycin and metformin in aging. Metformin and rapamycin promote longevity through distinct but complementary mechanisms. Rapamycin inhibits mTORC1, mimicking starvation and activating catabolic processes, such as mitophagy and mitochondrial biogenesis, via PGC-1α and SIRT1. Metformin, through partial complex I inhibition, lowers ATP levels and increases AMP, activating AMPK, which enhances NAD^+^ salvage pathways (via NAMPT) and promotes mitochondrial quality control. Metformin also shifts metabolism toward glycolysis by increasing lactate dehydrogenase (LDH) activity, thereby sustaining NAD^+^ levels. Additionally, both drugs inhibit mTORC1, reinforcing their shared role in energy regulation. Metformin reduces reactive oxygen species (ROS) formation, limiting lipid peroxidation and mitochondrial DNA (mtDNA) damage, thereby mitigating inflammaging. The synergistic effects of these drugs highlight their potential as longevity-promoting interventions.

**Figure 3 biomolecules-15-00614-f003:**
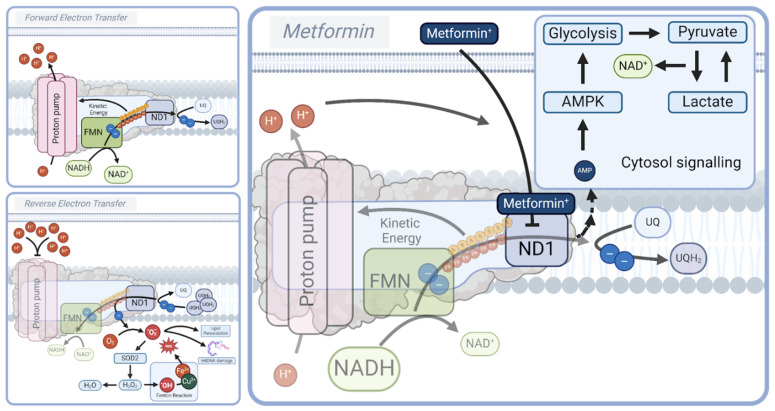
Metformin’s modulation of Complex I and mitochondrial metabolism. Metformin exerts its effects through the partial inhibition of the ND1 subunit of Complex I, preventing excessive electron backlogging and reducing reverse electron transfer (RET)-induced reactive oxygen species (ROS) production. This modulation lowers the proton motive force and ATP production, promoting glycolysis and increasing NAD^+^ levels via lactate dehydrogenase (LDH) activity. Elevated NAD^+^ enhances sirtuin activity, improving mitochondrial quality control and reducing inflammasome activation. Additionally, reduced ATP levels increase the AMP/ATP ratio, activating AMPK and promoting beneficial downstream effects, including autophagy, mitochondrial biogenesis (via PGC-1α), and antioxidant upregulation (via NRF2). Unlike potent Complex I inhibitors like rotenone, metformin’s self-limiting accumulation prevents excessive mitochondrial dysfunction, aligning with the hormesis theory by eliciting mild stress that enhances cellular resilience and longevity.

**Figure 4 biomolecules-15-00614-f004:**
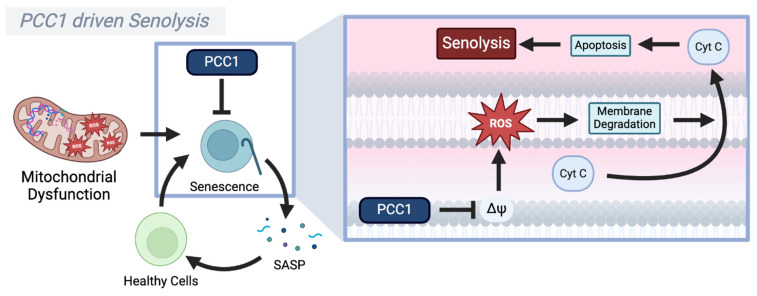
PCC1-induced senolysis through mitochondrial dysfunction. (**Left**) Mitochondrial dysfunction contributes to cellular senescence, leading to the secretion of senescence-associated secretory phenotype (SASP) factors, which further promote senescence in surrounding cells. PCC1, a senolytic compound, selectively targets senescent cells to induce apoptosis while reducing SASP-associated inflammation. (**Right**) PCC1 disrupts the mitochondrial membrane potential (ΔΨm), leading to excessive reactive oxygen species (ROS) production and mitochondrial membrane degradation. This results in cytochrome C (Cyt C) release, which triggers apoptosis and senolysis, effectively clearing the senescent cells. Through this mechanism, PCC1 may alleviate inflammaging and promote mitochondrial homeostasis.

**Figure 5 biomolecules-15-00614-f005:**
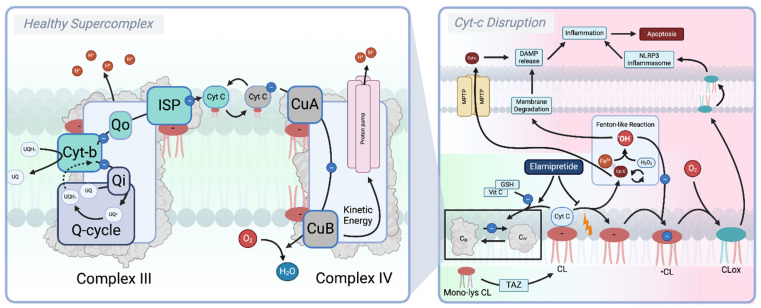
Respiratory Complex III–cytochrome C–Complex IV supercomplex in mitochondrial function and dysfunction. (**Left**) Under normal conditions, Complex III transfers electrons from reduced ubiquinol (UQH_2_) to cytochrome C (Cyt C), which, in turn, donates electrons to Complex IV, where oxygen (O_2_) is reduced to water (H_2_O). This process contributes to the proton-motive force (PMF) by translocating protons (H^+^) across the inner mitochondrial membrane (IMM). Cyt C is electrostatically associated with cardiolipin (CL), stabilizing the Complex III–Cyt C–Complex IV supercomplex and promoting efficient oxidative phosphorylation (OXPHOS). (**Right**) Mitochondrial dysfunction disrupts the Cyt C–CL interaction, leading to Cyt C unfolding and the exposure of its heme group, which facilitates peroxidase activity. This promotes lipid peroxidation, exacerbating mitochondrial damage and increasing reactive oxygen species (ROS) production. Oxidative stress and membrane degradation trigger inflammatory responses, including NLRP3 inflammasome activation and mitochondrial damage-associated molecular pattern (DAMP) release, contributing to inflammaging. Elamipretide, a cardiolipin-targeting peptide, stabilizes the Cyt C–CL interaction, thereby preventing peroxidase activity and supporting supercomplex integrity.

**Figure 6 biomolecules-15-00614-f006:**
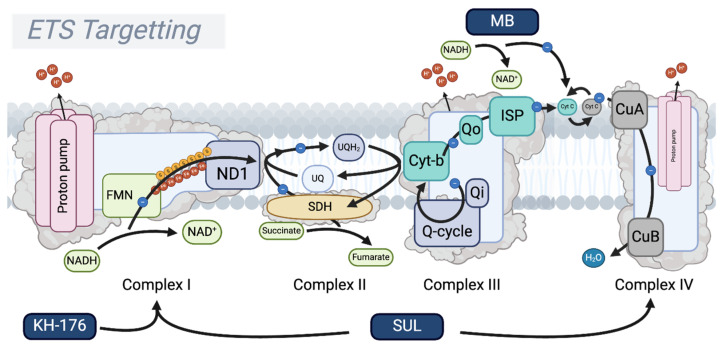
Pharmacological targeting of the electron transport system (ETS) to modulate mitochondrial function. The schematic depicts the mitochondrial ETS, highlighting key respiratory complexes (I–IV) and pharmacological interventions aimed at modulating oxidative phosphorylation. Electrons enter the chain via Complex I and Complex II, where ubiquinone (UQ) is reduced to ubiquinol (UQH_2_), and subsequently transfers electrons to Complex III. Cytochrome C (Cyt C) then facilitates electron transfer to Complex IV, where oxygen is reduced to water, driving ATP production. Methylene Blue (MB) acts as an alternative electron carrier by directly oxidizing NADH and reducing Cyt C, effectively bypassing Complex I and Complex III to prevent excessive ROS formation while enhancing oxidative phosphorylation. KH176 restores the mitochondrial redox balance by targeting Complex I, thus modulating oxidative stress through the thioredoxin/peroxiredoxin system. SUL compounds enhance the activity of both Complex I and the Complex III–Cyt C–Complex IV supercomplex, improving ATP production and preserving mitochondrial function under stress conditions. These pharmacological interventions hold promise for mitigating mitochondrial dysfunction in aging, neurodegenerative diseases, and ischemia–reperfusion injury.

## Data Availability

No new data were created in the preparation of this manuscript. All of the relevant literature is referenced below.

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
