# Peer review of "Targeting the Electron Transport System for Enhanced Longevity"

_biomolecules, 2025, doi:10.3390/biom15050614_

Round 1
Reviewer 1 Report
Comments and Suggestions for Authors
This is a very timely review on a very important topic that is overall well-written and informative. This is a deep dive into the topic but put together very well and the figures are very informative for the most part. To improve the overall appeal to a more general audience, Figure 1, should be edited to enhance the general reader's ability to understand the important concepts the authors are conveying. I section 3, the topic of RET depends on high PMF and conditions where this may exist should be explored more in-depth. For Figure 5 more rationale for its inclusion should be included, it seems a stand-alone to this reviewer. Finally, the authors might expand to include the use of mild uncouplers and mimics or at least touch on them given the goal of this review.
Author Response
Comment 1: “This is a very timely review on a very important topic that is overall well-written and informative. This is a deep dive into the topic but put together very well and the figures are very informative for the most part.”
Response 1: We thank the reviewer for the thoughtful and encouraging feedback. We’re glad to hear that they found the review timely and informative. Their comments are truly appreciated!
Comment 2: “To improve the overall appeal to a more general audience, Figure 1, should be edited to enhance the general reader's ability to understand the important concepts the authors are conveying.”
Response 2: We sincerely appreciate the reviewer’s suggestion to enhance the clarity of Figure 1. However, we feel that, when considered alongside the figure legend – as well as the context provided in the abstract and Section 1 – the figure already offers a clear and informative visual summary of the key concepts. That said, we are happy to consider minor adjustments should the reviewer feel specific elements could be further clarified
Comment 3: “I section 3, the topic of RET depends on high PMF and conditions where this may exist should be explored more in-depth.”
Response 3: We thank the reviewer for this thoughtful comment. To address it, we have expanded the text in Section 3 to more clearly describe conditions under which reverse electron transport (RET) occurs. Specifically, we now emphasize ischemia-reperfusion injury as a key context, highlighting how succinate accumulation during ischemia and its rapid oxidation during reperfusion drives RET through full reduction of the ubiquinone pool. We also note that this process is further exacerbated by high proton motive force, particularly under conditions of impaired or inhibited ATP synthase (Complex V) activity. These changes have been incorporated in lines 304–309, with additional references added to support this revision: This particularly evident in cases such as ischemia-reperfusion injury where succinate, accumulated during ischemia, is rapidly oxidized by Complex II upon reperfusion, fully reducing the UQ pool and driving RET at Complex I (Chouchani et al. 2014; Niatsetskaya et al. 2012; Sahni et al. 2018). This process is further exacerbated by high proton motor force which can result from impaired or inhibited Complex V (ATP -synthase) activity (Chouchani et al. 2014).
Comment 4: “For Figure 5 more rationale for its inclusion should be included, it seems a stand-alone to this reviewer.”
Response 4: We thank the reviewer for this valuable observation. Figure 5 was originally intended as a visual aid to support the introduction of the concept of the mitochondrial supercomplex. After revisiting both the comment and the corresponding section of the text, we agree that the figure appeared somewhat standalone in its initial placement. To improve coherence and better align the figure with the accompanying explanation, we have moved Figure 5 from Section 5.1 to Section 5.2, where the concept is fully developed. We believe this adjustment strengthens the overall flow and clarity of the manuscript.
Comment 5: “Finally, the authors might expand to include the use of mild uncouplers and mimics or at least touch on them given the goal of this review.”
Response 5: We thank the reviewer for this thoughtful suggestion. While we chose not to provide an exhaustive overview, we believe the manuscript does touch upon several relevant examples of uncouplers and mimetics. For instance, we discuss MitoTEMPO as a mitochondrial-targeted superoxide dismutase (SOD) mimic, Metformin as a compound with mild uncoupling properties, and PCC1 as a strong uncoupler used in the context of senolytic therapy. Our aim was to offer a representative and conceptually diverse overview of compounds that target mitochondrial pathways to promote longevity. A more comprehensive discussion of uncouplers and mimetics, while certainly valuable, is beyond the scope of this review. We hope the reviewer agrees that the examples provided offer a meaningful framework and set the stage for future research. That said, if the reviewer has a specific uncoupler or reference in mind, we would be happy to consider including it.
Reviewer 2 Report
Comments and Suggestions for Authors
In the manuscript titled "Targeting the Electron Transport System for Enhanced Longevity," the authors highlight the pivotal role of mitochondrial dysfunction in the ageing process. They propose that mitochondrial DNA damage may serve as a key driver of ageing. Furthermore, the manuscript reviews a wide array of compounds that modulate mitochondrial function, examining their impact on mitochondrial homeostasis and organismal longevity. Ultimately, the authors propose a pharmacological strategy aimed at targeting mitochondrial pathways to potentially achieve anti-ageing effects.
This is a well-structured and informative manuscript that provides valuable insights for the scientific community focused on mitochondria and age-related research. However, I have a few minor concerns:
1)The authors propose that mitochondrial DNA (mtDNA) damage may act as an initial trigger in the ageing process. However, the manuscript would benefit from a clearer explanation of the underlying mechanisms leading to mtDNA damage.
2)Authors write in line 389:"Currently, there are no data available to discuss the use of NR or NMN in model organisms beyond yeast to confer longevity [153]–[155]". Please add the below study from Fang et al.,2019 that provide data both in C elegans and Drosophila
Fang, E.F., Hou, Y., Lautrup, S. et al. NAD+ augmentation restores mitophagy and limits accelerated aging in Werner syndrome. Nat Commun 10, 5284 (2019). https://doi.org/10.1038/s41467-019-13172-8
3) Authors said (Lines 153-154)that: " Concurrently, the removal of damaged mitochondria through mitophagy declines with age [44]. This is largely due to a reduction in PINK1/Parkin signaling...". Since mitophagy can also be activated by a PINK/Parkin independent pathway, please also provide information for this signalling during aging
4)In line 426 replace "Given thatantioxidants ..." to "Given that antioxidants..."
Author Response
Comment 1: “This is a well-structured and informative manuscript that provides valuable insights for the scientific community focused on mitochondria and age-related research.”
Response 1: We thank the reviewer for the thoughtful and encouraging feedback. We’re glad to hear that they found the review well-structured and informative.
Comment 2: “The authors propose that mitochondrial DNA (mtDNA) damage may act as an initial trigger in the ageing process. However, the manuscript would benefit from a clearer explanation of the underlying mechanisms leading to mtDNA damage.”
Response 2: We thank the reviewer for raising this point. However, we believe that the manuscript already provides a detailed explanation of the mechanisms underlying mtDNA damage and its relevance to aging, particularly in Section 1.2. Here, we outline how mitochondria generate ROS as a byproduct of electron transfer through the ETS, and how proximity to the radical-producing complexes, the lack of protective histones, and limited repair capacity render mtDNA especially vulnerable to oxidative damage. We further explain how this damage compromises OXPHOS function, initiating a self-reinforcing cycle of energy deficit and additional ROS production, leading to clonal expansion of mutated mtDNA, disrupted proteostasis, and ultimately mitochondrial and cellular dysfunction. These mechanisms are also visually summarized in Figure 1. Given the focus of this review on mitochondrial therapeutic targets for aging, we aimed to provide a solid mechanistic foundation for mtDNA damage without delving into a more exhaustive molecular discussion—which, while certainly valuable, is beyond the scope of the present work. We hope the reviewer can agree that the current level of detail sufficiently supports our central premise—that mitochondrial dysfunction is an initiating and propagating factor in aging.
Comment 3: “Authors write in line 389:"Currently, there are no data available to discuss the use of NR or NMN in model organisms beyond yeast to confer longevity [153]–[155]". Please add the below study from Fang et al.,2019 that provide data both in C elegans and Drosophila
Fang, E.F., Hou, Y., Lautrup, S. et al. NAD+ augmentation restores mitophagy and limits accelerated aging in Werner syndrome. Nat Commun 10, 5284 (2019). https://doi.org/10.1038/s41467-019-13172-8.”
Response 3: We thank the reviewer for this helpful suggestion. We have now incorporated the study by Fang et al. (2019), which demonstrates that NAD⁺ augmentation restores mitophagy and extends lifespan in models of Werner syndrome in both C. elegans and Drosophila. This reference directly supports the potential of NAD⁺ repletion to counteract progeroid features and provides important evidence beyond yeast models. The change has been implemented in lines 390 and 396, and the reference has been added to the manuscript: However, it is worth mentioning that NAD+ augmentation has been shown to extend lifespan and restore mitophagy in models of Werner syndrome – an accelerated aging disorder – in both C. elegans and Drosophila, suggesting that NAD+ repletion does counteract specific progeroid phenotypes and may prove effective against aging in future studies (Fang et al. 2019). It is pertinent to highlight that studies disagree on whether aging is actually associated with reduced levels of NAD+ [147].
Comment 4: “ Authors said (Lines 153-154)that: " Concurrently, the removal of damaged mitochondria through mitophagy declines with age [44]. This is largely due to a reduction in PINK1/Parkin signaling...". Since mitophagy can also be activated by a PINK/Parkin independent pathway, please also provide information for this signalling during aging.”
Response 4: We appreciate the reviewer’s insightful comment. We fully agree that mitophagy can be activated through PINK1/Parkin-independent pathways as well. However, our intent in this section was not to provide an exhaustive overview of mitophagy mechanisms, but rather to highlight the age-related decline in mitophagy capacity – particularly through the well-characterized PINK1/Parkin axis. Given the tight link between mitochondrial quality control and aging, we felt this focus was appropriate to support the broader narrative. That said, if the reviewer has a specific PINK1/Parkin-independent pathway in mind that would meaningfully enhance this section, we would be happy to consider including it.
Comment 5: “In line 426 replace "Given thatantioxidants ..." to "Given that antioxidants...".”
Response 5: The missing space in line 426 was added.
Round 2
Reviewer 1 Report
Comments and Suggestions for Authors
The authors have improved the review and addressed the bulk of my comments.
Reviewer 2 Report
Comments and Suggestions for Authors
I would like to thank the authors for their response. Iagree to publish the manuscript in its current condition.